Evaluation of granular anaerobic ammonium oxidation process for the disposal of pre-treated swine manure

Ni Shou-Qing sqni@sdu.edu.cn
Yang Ning
Shandong Provincial Key Laboratory of Water Pollution Control and Resource Reuse, School of Environmental Science and Engineering, Shandong University , Jinan , China
Liu Yu
Electronic publication date: 2014 Apr 8
Publication date: 2014
Volume: 2
Electronic Location ID: e336
Received 2013 Dec 31; Accepted 2014 Mar 17
Copyright: © 2014 Ni and Yang
Copyright year: 2014
Copyright holder: Ni and Yang
License: This is an open access article distributed under the terms of the Creative Commons Attribution License, which permits unrestricted use, distribution, and reproduction in any medium, provided the original author and source are credited.
License URL: https://creativecommons.org/licenses/by/3.0/

Keywords: Granular sludge, Anammox, Swine manure, Mass balance, Organic matter, Nitrogen

Funding: The National Natural Science Foundation of China 51108251 & 21177075 Research Award Fund for Outstanding Middle-aged and Young Scientist of Shandong Province BS2012HZ007 Independent Innovation Foundation of Shandong University 2012GN001 Overseas Personnel Pioneer Plan of Jinan 20110406 The National Natural Science Foundation of China (Nos. 51108251 & 21177075), Research Award Fund for Outstanding Middle-aged and Young Scientist of Shandong Province (No. BS2012HZ007), Independent Innovation Foundation of Shandong University (No. 2012GN001) and the Overseas Personnel Pioneer Plan of Jinan (No. 20110406). The funders had no role in study design, data collection and analysis, decision to publish, or preparation of the manuscript.

==============================
With rising environmental concerns on potable water safety and eutrophication, increased media attention and tighter environmental regulations, managing animal waste in an environmentally responsible and economically feasible way can be a challenge. In this study, the possibility of using granular anammox process for ammonia removal from swine waste treatment water was investigated. A rapid decrease of NO2−–N and NH4+–N was observed during incubation with wastewater from an activated sludge deodorization reactor and anaerobic digestion-partial oxidation treatment process treating swine manure and its corresponding control artificial wastewaters. Ammonium removal dropped from 98.0 ± 0.6% to 66.9 ± 2.7% and nearly absent when the organic load in the feeding increased from 232 mg COD/L to 1160 mg COD/L and 2320 mg COD/L. The presence of organic carbon had limited effect on nitrite and total nitrogen removal. At a COD to N ratio of 0.9, COD inhibitory organic load threshold concentration was 727 mg COD/L. Mass balance indicated that denitrifiers played an important role in nitrite, nitrate and organic carbon removal. These results demonstrated that anammox system had the potential to effectively treat swine manure that can achieve high nitrogen standards at reduced costs.

Introduction

Large concentrated swine feeding operations throughout the world are presently producing a huge amount of manure with abundant nitrogen and phosphorus as well as organic matter (Zhang et al., 2006). Liquid swine manure can provide essential nutrients for plant growth. On the other hand, continuing land application for manure disposal could result in excessive nutrient loss from soil to water, causing eutrophication that deteriorates water quality (Karlen, Cambardella & Kanwar, 2004). Manure also contributes to the production of greenhouse gas emissions (Thorman et al., 2007). Usually, effluent from anaerobic wastewater treatment processes is characterized by a high concentration of nitrogen and a low concentration of organic matters (i.e., a low C/N ratio) (Kataoka et al., 2002). Biological nitrogen removal is achieved mostly by complete oxidation to NO3− with surplus oxygen and subsequent reduction of NO3− to N2 gas under anoxic conditions at the expense of COD. If the C/N ratio in wastewater is low, additional carbon for denitrification is required. Special attention needs to be given to N2O gas emissions during biological nitrogen removal process (Hu et al., 2013; Kong et al., 2013). Therefore, there is an urgent call for development of sustainable technologies for removals of N from swine manure with respect to environmental and agricultural benefits.

As an autotrophic and cost-effective process for nitrogen removal, anaerobic ammonium oxidation (anammox) process is an alternative to the traditional biological nitrification–denitrification process (Ni & Zhang, 2013). The discovery of anammox process brought significant progress to conventional biological nitrogen removal process. Some promising characteristics make the anammox process an attractive and sustainable option (Abma et al., 2007), such as low excess sludge production, no need for aeration and no addition of biodegradable organic carbon (Ni et al., 2010a). In comparison to the nitrification–denitrification process, anammox consumes 100% less biodegradable organic carbon and at least 50% less oxygen (Tal, Watts & Schreier, 2006).

A long start-up period is expected in the anammox process due to the slow growth rate of anammox bacteria (Strous et al., 1998). Reducing the potential for anammox sludge wash-out from the reactor becomes an effective strategy to shorten the start-up period of the anammox process. Thus, different types of reactors have been adopted to meet this goal including the continuous stirred-tank reactor, anaerobic biological filtrated reactor, sequencing batch reactor (SBR), up-flow reactor and biofilm reactor (Imajo, Tokutomi & Furukawa, 2004; Isaka, Sumino & Tsuneda, 2007; Strous et al., 1998; van Dongen, Jetten & van Loosdrecht, 2001). Faster growth of anammox bacteria was observed in a membrane bioreactor with a doubling time of less than 10 days, leading to a purity of 97.6% (van der Star et al., 2008). The formation of sludge aggregates was reported to keep a large amount of active anammox biomass from washing out of the reactor (Imajo, Tokutomi & Furukawa, 2004). Therefore, granulation is a feasible method for anammox enrichment.

Only a few studies have investigated the possibility of using the anammox process for ammonia removal from swine waste treatment water (Ahn, Hwang & Min, 2004; Hwang et al., 2005; Molinuevo et al., 2009; Waki et al., 2007). However, there is still a big gap regarding the performance of anammox granules for the treatment of swine manure. The objective of this study was to develop a potential swine manure treatment process that can achieve high nitrogen standards at reduced costs by investigating the performance of anammox granular process fed with pre-treated swine manure effluent.

Materials & Methods

Granules cultivation and reactor operation

Two lab-scale up-flow anaerobic sludge blanket (UASB) reactors were inoculated with 900 mL anammox granules from a running UASB reactor (Ni et al., 2011). The mixed liquor suspended solid and mixed liquor volatile suspended solid of the seed sludge were 4.24 g/L and 3.35 g/L, respectively. The reactors were running in a continuous mode at an HRT of approximately 1.0 days. The effluent was recycled from the bottom of the reactor. One reactor was designated as the control.

The reactors were operated at 35 °C with a working volume of 3.0 L. Different sizes of gravel were placed in the bottom of the reactors. The pH in the reactor was controlled approximately 7.5 using CO2 purge and the anoxic condition was created via argon gas. Before feeding with swine manure, the reactor was pumped with synthetic wastewater prepared by adding ammonium and nitrite to a mineral medium in the required amounts in the form of (NH4)2SO4 and NaNO2. The composition of the mineral medium was (g/L): KHCO3 0.5, KH2PO4 0.0272, MgSO4 ⋅ 7H2O 0.3, CaCl2 ⋅ 2H2O 0.18 and 1 mL trace elements solutions I and II (Ni et al., 2010a). The synthetic wastewater was deoxygenated by flushing with argon gas before feeding to the reactor.

The effluents from an activated sludge deodorization reactor and anaerobic digestion-partial oxidation treatment (AD-PO) process treating swine manure were collected. The effluent from the activated sludge deodorization reactor contained 220 mg/L NH4+–N, 265 mg/L NO2−–N, 125 mg/L NO3−–N, and 230 mg/L COD. The effluent from the AD-PO process contained 610 mg/L NH4+–N, 650 mg/L NO2−–N, 1350 mg/L NO3−–N, and 2320 mg/L COD. Both reactors were initially fed with synthetic wastewater for 35 days. Then one reactor was fed with the effluent from the activated sludge deodorization reactor without dilution and the other one with the effluent from the AD-PO process, which was done gradually in increments of 10%, 20%, 50% and 100% (v/v).

EPSs extraction and analysis

The EPSs in the granules were extracted using cation exchange resin (CER). In general, sludge samples were harvested by centrifugation at 3000 rpm for 15 min at 4 °C and then the sludge pellets were re-suspended in phosphate buffer solution (pH 7.0) and the solution was transferred to an extraction bottle, followed by the CER addition with a dosage of 75 g/g suspended solids. These suspensions were stirred at 600 rpm at 4 °C for 2 h. After removing the settled CER, the solutions were centrifuged at 8000 rpm for 30 min to remove remaining sludge components. The supernatants were then filtered through 0.45 µm cellulose membranes and used as the EPSs fraction for protein and carbohydrate analyses. The protein content in the EPSs was determined according to the Bradford protein assay with bovine serum albumin as the standard (Bradford, 1976). The carbohydrate content in the EPSs was measured using the Anthrone method with glucose as the standard (Gaudy, 1962). The total EPSs content was measured as the sum of these two substances.

DNA extraction and quantitative real-time polymerase chain reaction (PCR)

Total genomic DNA was extracted by the modified 2% cetyl trimethyl ammonium bromide-based protocol (Allen et al., 2006). Genomic DNA preparation was determined with an ND-1000 NanoDrop spectrophotometer (NanoDrop Technologies, Wilmington, DE, USA) and purified DNA samples were stored in sterile deionized water at −20 °C until used. Quantitative PCR was then processed based on the description of literature (Ni et al., 2011).

Fluorescence in situ hybridization

Fluorescence in situ hybridizations (FISH) and 4,6-diamidino-2-phenylindole (DAPI) staining were performed according to the procedure described by Amann, Krumholz & Stahl (1990) and Sun et al. (2009). The 16S rRNA-targeted oligonucleotide probes used in this study were AMX368F (CCTTTCGGGCATTGCGAA) and DEN220 (GGCCGCTCCGTCCGC) for anammox and denitrifying bacteria. Images were acquired using an epifluorescence microscope (Olympus BX51; Olympus Optical, Tokyo, Japan) together with the standard software package delivered with the instrument (version 4.0). The images were taken by an Olympus U-CMAD 3 camera (Olympus Optical, Tokyo, Japan).

Analysis

Ammonia was measured by selective electrode according to the Standard Methods (APHA, AWWA & WEF, 1998). Nitrite and nitrate concentrations were determined by ion-chromatography (DX 500; Dionex, USA). The measurement of COD was carried out according to the Standard Methods 5220 (APHA, AWWA & WEF, 1998). The SS and VSS were determined by the weighing method after being dried at 103–105 °C and burnt to ash at 550 °C (APHA, AWWA & WEF, 1998). For the electron microscopy observation, samples were fixed with 2% paraformaldehyde and 2% glutaraldehyde in 0.1 M cacodylate buffer at 4 °C for 24 h. Samples were then prepared following the method of Ni et al. (2011). For the transmission electron microscopy (TEM), images were captured using a JEM 2100 200 kV scanning and transmission electron microscope (Japan Electron Optic Laboratories, Peabody, MA). For the scanning electron microscopy (SEM), morphology characteristics of the biomass specimens were observed using a JEOL 5800LV SEM (JEOL, Peabody, MA).

Results and Discussion

Control reactor performance and characteristics of anammox granules

Feeding with synthetic wastewater, the control experiment was carried out at an HRT of 1.0 days and the influent NH4+–N to NO2−–N ratio was kept at around 1.0. Stable performance was realized in several days after the addition of anammox granules. The reactor was run for 35 days with high substrate removal. The average effluent ammonia and nitrite concentrations were 1.0 ± 0.4 and 0.6 ± 0.9 mg N/L, respectively (Fig. 1), leading to the ammonia and nitrite removal efficiencies of 98.0 ± 0.8% and 98.9 ± 1.7%. Due to the production of nitrate by the anammox process, the total nitrogen (TN) removal efficiency was only 83.6 ± 1.1%.

Figure 1 Nitrogen removal performance of control reactor feeding with synthetic wastewater.

During the experiment, the granules were sampled for the microscope observation. As shown in Fig. 2A, the sludge in the reactor was reddish, semitransparent and easily formed granules. Each part of the granules was densely incorporated with others, which favored the granules joining tightly and existing stably. This structure was possibly formed due to the shear forces of the effluent recirculation currents (Ni et al., 2011). Spherical shaped bacteria, which were supposed to be anammox bacteria (Jetten et al., 1999), were observed (Fig. 2B). Transmission electron micrograph shows that anammox bacterial cells have an irregular morphology (Fig. 2C). In this paper, the cells displayed an identical pattern of organization to other anammox species (Kartal et al., 2008).

Figure 2 Images of anammox bacteria.

(A) Image showing the reddish anammox granules in a beaker. (B) Scanning electron micrograph showing anammox bacteria surrounded by bacterial extracellular polymeric substances (bar = 400 nm). (C) Transmission electron micrograph showing anammox bacterial cells (bar = 500 nm).

From the SEM image (Fig. 2B), anammox cells were surrounded by bacterial extracellular polymeric substances (EPS). EPS were believed to play a fundamental role during the formation of anammox granules (Ni et al., 2010b). Generally, bacterial EPS, consisting of polysaccharides, proteins, nucleic acids, and lipids, are sticky materials secreted by microorganisms, acting as cementing substances in biofilms and flocs (Characklis & Marshall, 1990; Frolund et al., 1996). Proteins and carbohydrates were reported to be the dominant components in the extracted EPS and therefore were usually employed to represent the EPS content. During the experiment, the proteins and carbohydrates contents in the extracted EPSs of the granules were analyzed. The total EPSs content was measured as the sum of these two substances. The proteins and carbohydrates in anammox granules were 56.7 ± 2.8 and 65.7 ± 3.2 mg/g VSS with a protein/carbohydrate (PN/PS) ratio of approximately 0.9. The total EPSs contents in anaerobic and aerobic granules were around 60 mg/g VSS (Wu et al., 2009; Zheng & Yu, 2007), substantially lower than that for the anammox granules (total EPSs content was about 122.4 mg/g VSS) in this study. The PN/PS ratios were higher than 2.0 for anaerobic, aerobic and nitrifying granules (Martinez et al., 2004; Wu et al., 2009; Zheng & Yu, 2007), while it was lower than 1.0 for denitrifying granules (Bhatti et al., 2001), similar to that of this study. This suggested that proteins might be the key EPS constituents for anaerobic, aerobic, and nitrifying granules, but carbohydrates might play a significant function in the development of denitrifying and anammox granules.

Quantitative real-time PCR analysis was used to quantify the microbial community of the granules in the reactor, using the assay based on the 16S rRNA gene-specific set of primers AMX809F/AMX1066R. The data indicated that anammox bacteria comprised about 91% cells in the microorganisms’ community, resulting in high NH4+–N and NO2−–N removal efficiencies. FISH images also showed that anammox bacteria constituted the majority of cells in the community (Fig. 3). As shown in Fig. 3, a small amount of denitrifying bacteria was observed to be existed together with anammox microorganisms.

Figure 3 FISH image.

Identification of microorganisms by hybridizing with different fluorescent-labeled probes. (A) Probe AMX368F targeting for anammox bacteria. (B) Probe DEN220 targeting for denitrifying bacteria.

Nitrogen removal from pretreated swine manure

After more than a month stable operation, ammonium and nitrite removal rates in both reactors reached over 95%, demonstrating that both anammox granular reactors were ready for further study. Reactor I was fed with the effluent, which contained 220 mg/L NH4+–N, 265 mg/L NO2−–N, 125 mg/L NO3−–N and 230 mg/L COD, from the activated sludge deodorization reactor for about 50 days. As shown in Fig. 4, during the late 22 days, NH4+–N, NO2−–N, and TN removal rate were 92.2 ± 1.5%, 99.3 ± 0.9%, and 72.0 ± 1.4%, respectively. Ammonium and nitrite removal rates were very high, indicating the good activities of anammox microorganisms. Due to the existence of NO3−–N in the feeding, the calculated TN removal rate ([removed NH4+–N + NO2−–N + NO3−–N]/ [influent NH4+–N + NO2−–N + NO3−–N]) was only 72.0%. The traditional biological nutrient removal may realize higher TN removal with much higher costs (Kunz, Miele & Steinmetz, 2009). An 8 h per cycle sequencing batch reactor with alternating anaerobic-anoxic-anoxic/anaerobic-anoxic/aerobic conditions realized 95% of TN reductions for swine manure treatment (Zhang et al., 2006). Besides anammox bacteria, other species such as denitrifiers may contribute to nitrogen removal from wastewater. Process stoichiometry was calculated to get a deep insight of their relations (Fig. 4). The stoichiometry molar ratios of NO2−–N to NH4+–N conversion and NO3−–N production to NH4+–N conversion were 1.30 ± 0.01 and 0.14 ± 0.008. More nitrite was removed and fewer nitrates were produced. This finding indicated that organic matters enhanced the nitrogen removal by favoring the denitrifiers and they consumed the surplus nitrite and produced nitrate (Eqs. (1) and (2)) (Ni et al., 2012). Anammox may also contribute to more nitrite removal and less nitrate production. Kartal et al. (2007) indicated that anammox bacteria could be mediating dissimilatory nitrate reduction to ammonium, followed by the anaerobic oxidation of the produced ammonium and nitrite with the overall end-product of dinitrogen gas. Though anammox bacteria were disguised as denitrifiers, its pathway was different from ‘classical’ denitrification pathway, in which N2 is produced via nitrite, nitric oxide and nitrous oxide. In anammox pathway, nitrate was reduced to dinitrogen gas via nitrite and ammonium. This process was relatively slow. The nitrate reduction to nitrite proceeded at a rate of 0.3 ± 0.02 fmol/cell/day which was only 10% of the anammox rate. So the surplus nitrite and produced nitrate were mostly consumed by denitrifiers. (1) NO3−+0.29CH3CH2CH2COOH+H2CO3→0.034C5H7O2N+HCO3−+1.54H2O+0.986CO2+0.483N2

(2) NO2−+0.19CH3CH2CH2COOH+H2CO3→0.037C5H7O2N+HCO3−+1.14H2O+0.585CO2+0.48N2.

Figure 4 Treating the effluent after activated sludge deodorization reactor.

NH4+–N, NO2−–N and TN removal (left axis) and process stoichiometry (right axis) during implementation of the effluent after activated sludge deodorization reactor.

Reactor II was fed with the effluent, which contained 610 mg/L NH4+–N, 650 mg/L NO2−–N, 1350 mg/L NO3−–N, and 2320 mg/L COD, from the AD-PO process for about 2 months. The presence of organic matters was found to affect anammox process adversely (van de Graaf et al., 1996). Anammox microorganisms could not compete with denitrifiers for nitrite and may result in complete inactivation of anammox activity under high organic matter concentration (Güven et al., 2005; Molinuevo et al., 2009). So the feeding in reactor II was done gradually in increments of 10%, 20%, 50% and 100% (Fig. 5). The addition of 10% of AD-PO effluent (organic load of 232 mg COD/L) resulted in up to 98.0 ± 0.6% of high ammonium removal, compared with ammonium removal for activated sludge deodorization reactor effluent (92.2 ± 1.5%, organic load of 230 mg COD/L). The difference was caused by higher ammonium concentration (∼220 mg N/L) of activated sludge deodorization reactor effluent than that (∼60 mg N/L) after AD-PO treatment. As high levels of free ammonia are toxic to the anammox process (Waki et al., 2007), pretreatments, for example partial oxidation of ammonia to oxidized nitrogen, may facilitate anammox reaction. Then, the feeding rate was increased gradually to organic loads of 464, 1160, and 2320 mg COD/L, corresponding to 20%, 50% and 100% of AD-PO effluent. Ammonium removal rates were 88.0 ± 1.0% and 66.9 ± 2.7% when 464 and 1160 mg COD/L were pumped into the reactor. When 2320 mg COD/L was added, the ammonium removal was dropped quickly to nearly absent. Similar phenomena were reported in literature. Though high ammonium removal of 92.1 ± 4.9% was achieved for 2% (v/v) UASB-post-digested pig manure effluent (95 mg COD/L, 75.6 mg NH4+–N) by using anammox process, ammonium removal fell to 0% when 5% (v/v) UASB-post-digested effluent was added (237 mg COD/L, 189 mg NH4+–N) (Molinuevo et al., 2009). If coupled with nitrification process, anammox could treat many types of high strength industrial wastewaters (Daverey et al., 2013; Jenni et al., 2014; Lackner et al., 2014; Li et al., 2014). This study indicated that anammox could also treat swine manure with high organic loading rate.

Figure 5 Treating the effluent after anaerobic digestion-partial oxidation treatment.

NH4+–N, NO2−–N and TN removal (left axis) and process stoichiometry (right axis) during gradual implementation of the effluent after anaerobic digestion-partial oxidation treatment.

However, nitrite removal was seldom affected by organic loading rate. Most of the time, over 95% of nitrite removal was achieved (Fig. 5). The calculated TN removal rate was less than 50% due to the high concentration of NO3−–N in the feeding. The stoichiometry molar ratios of NO2−–N to NH4+–N conversion were 1.07 ± 0.01 and 1.18 ± 0.01 when 232 and 464 mg COD/L were applied, close to the theoretical value (Strous et al., 1998). And this value increased to 1.58 ± 0.07 at 1160 mg COD/L. When organic load of 2320 mg COD/L was achieved, ratios of 3.82–8.39 were obtained and as time went by, this ratio increased up to 44.0. In this case the heterotrophic denitrification was the major reaction involved in ammonium removal (Molinuevo et al., 2009). Results from the mass balance showed that the participation of anammox process in the total ammonium and nitrite removal decreased when a high percent of AD-PO effluent was implemented, which was replaced by the denitrification.

The physiological changes of biomass were also observed. When organic load of 2320 mg COD/L was achieved, the disintegration of biomass was registered. The red granules began to turn black and more aggregated biomass disassembled to small parts. Due to the change of running conditions by the addition of more organic matters, anammox communities decreased and denitrifiers took charge of ammonium removal eventually. In this situation, slowly growing anammox bacteria (Y = 0.066 ± 0.01) are incapable of competing with denitrifiers with higher growth yield (Y = 0.3). FISH images also showed that there was a reduction in the number of anammox cells when 2320 mg COD/L was added in comparison with the abundance of anammox microorganisms at organic load of 232 mg COD/L.

Effect of organic matters on anammox performance

Literature review showed that high content of organic matter usually inhibited anammox activity. In this study, at a COD to N ratio of 0.9, COD inhibitory organic load threshold concentration was 727 mg COD/L (Fig. 6). Previously, we found the threshold was 308 mg COD/L and 3.1 for COD to N ratio (Ni et al., 2012). Both organic matter concentrations and COD to N ratios affect the performance of anammox bacteria without a general agreement (Chamchoi, Nitisorvut & Schmidt, 2008; Molinuevo et al., 2009). Batch tests showed that 25 and 50 mM acetate resulted in 22 and 70% inhibition in anammox process (Dapena-Mora et al., 2007). Güven et al. (2005) indicated that even 0.5 mM of methanol resulted in the immediate and complete inactivation of anammox activity. About 300 mg COD/L (COD to N ratio of 2) was found to inactivate or eradicate anammox communities under concurrent operation of anammox and denitrification (Chamchoi, Nitisorvut & Schmidt, 2008). At a COD to N ratio of 0.5, COD inhibitory organic load threshold concentration (defined when ammonia removal dropped to 80%) was 142 and 242 mg/L when treating different wastewaters (Molinuevo et al., 2009). High free ammonia (FA) concentration had a negative effect on anammox process (Kim et al., 2009; Waki et al., 2007). Ni & Meng (2011) found that 27 mg/L FA could cause more than 50% bacterial activity loss, and as high as 61–63 mg/L FA would totally inhibit granular anammox microorganisms. During this experiment, FA concentration was lower than 5.0 mg/L, much lower than the possible toxic concentration to anammox biomass.

Figure 6 Effect of organic matter on anammox performance treating pretreated swine manure.

To further understand the effect of organic matters on anammox performance, mass balance evaluation of participation of different processes was done as illustrated in Table 1. At low COD to N ratios, variation of COD to N ratio had limited effect on anammox performance. At a COD to N ratio of 0.9, anammox accounted for 98.9% nitrite removal, while at a COD to N ratio of 0.4, anammox accounted for 88.5% nitrite removal (Table 1). The difference was mainly caused by influent substrate concentration. Somehow, COD removal in reactor I had higher efficiency, i.e., over 70%. Less than 50% COD was removed in reactor II at all conditions. COD removal was resulted from denitrification by the denitrifying communities mainly using nitrite as electron acceptor. Hence the competition for nitrite as electron acceptor between the denitrifying bacteria and the anammox communities existed in the reactors. Nitrite (ammonia) consumption via anammox was 98.9% (98.0%) when influent COD and ammonium concentrations were 232 mg/L and 67 mg N/L, but it decreased sharply to 27.5% (<5%) at 2320 mg COD/L and 615 NH4+–N/L (Table 1), indicating that denitrification prevailed anammox process gradually. Denitrification helped to remove nitrite and nitrate when organic matter was available and will become the dominant route in the reactor in time.

Table 1 Mass balance evaluation of participation of different processes.

COD conc.
(mg/L)	COD
to N ratio	NH4+–N
removal
(mg N/L)	NO2−–N removal
(mg N/L)	NO3−–N
(mg N/L)	COD
removal (mg/L)	Inf. NH4+–N	NO2−–N
consumption via anammox
(%)	
			Anammox	Denitrification	Productiona	Removalb	Finalc				
0	—	48.8	51.0	—	15.8	—	15.8	—	50.0	100	
232.0	0.9	65.5	68.8	0.8	21.0	11.8	9.2	41.8	67.0	98.9	
464.0	0.9	115.3	121.1	15.4	37.4	15.7	21.7	86.6	121.1	88.7	
1160.0	0.9	211.9	222.5	107.1	68.7	40.6	28.1	369.2	304.5	67.5	
2320.0	0.9	166.9	175.2	462.6	54.1	29.0	25.1	1097.6	615.0	27.5	
230.0	0.4	211.2	221.8	28.9	68.4	32.5	35.9	173.1	231.0	88.5	
Notes.

a Values denote nitrate produced via anammox process.

b Values means nitrate removed by denitrification process.

c Values stand for final concentration of effluent nitrate.

Conclusion

Nitrogen removal has become a major focus in swine manure treatment since nitrogen is the nutrient concerning the application amount of the manure produced in accordance with an increasing number of governmental regulations. As a novel, autotrophic and cost-effective alternative to the traditional biological nitrification/denitrification removal process, anammox process was proved to be effective for swine manure nitrogen removal. With increasing organic matters, ammonium removal via anammox decreased and the role of denitrifiers in nitrite, nitrate and COD removal became significant, proved by mass balance. To facilitate anammox performance, effective pre-treatment to reduce influent organic carbon was necessary. The introduction of organic matters favored the growth of denitrifiers. At low COD to N ratios, variation of COD to N ratios had limited effect on anammox performance.

Additional Information and Declarations

Competing Interests

Author Contributions

The authors declare there are no competing interests.

Shou-Qing Ni conceived and designed the experiments, performed the experiments, analyzed the data, contributed reagents/materials/analysis tools, wrote the paper, prepared figures and/or tables, reviewed drafts of the paper.

Ning Yang performed the experiments, wrote the paper, prepared figures and/or tables, reviewed drafts of the paper.

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
