# Peer review of "Evaluation of granular anaerobic ammonium oxidation process for the disposal of pre-treated swine manure"

_PeerJ, doi:10.7717/peerj.336_

## Round 0.1 · original submission · Minor Revisions

I noticed that several long sentences in this manuscript were taken from the authors' previous publications. Thus, they should be amended in a proper manner.

Reviewer 1 ·

Basic reporting

No Comments

Experimental design

No Comments

Validity of the findings

No Comments

Additional comments

This article applies granular anammox process for removing ammonia in swine manure. The highest ammonium removal rate can be upon 98%. However, increasing the concentration of COD until 2320 mg COD/L caused lower ammonium removal rate (66.9%). Furthermore, results indicated that denitrifying bacteria may help to remove nitrite, nitrate and organic carbon. This study proofed that anammox can effectively treat swine manure. There are some comments that may further enhance the quality of this article.
1. There are some mistakes in English. Authors can ask native English speaker to revise current manuscript.
2. Although this field may be a relatively new topic, to compare the performance in this study with other literatures can enhance the comprehension of audience. Authors are also encouraged to cite more recent papers related to various wastewaters treated by anammox process.
3. Since the results indicated that denitrifying bacteria may help to remove nitrite, authors should provide some direct evidences such as image of electrophoresis loading samples from reactions of polymerase chain reaction, which using denitrified specific primers.

Reviewer 2 ·

Basic reporting

This paper presnts the results on the possibility of using granular anammox process for ammonia removal from swine waste. Also, the study focused on the effects of COD/N ratio on the performance of the process. Both are meaningful to application and research.

Experimental design

Logic.

Validity of the findings

Data suppots the findings and major conclusions.

Additional comments

May need to notice on the aspects below:
The effects of C/N on the process is complex, may involve the facts i. NH3, which relates to NH4 concentration; and ii. the denitrification functions which some anammox species can play. It might be helpful to discuss these points in the disccsion part.

In fact according to the findings presented high COD makes the current process to treat swine waste low efficiecny. A improved pre-treatment to low down feed COD may be necessary. It may need to say in the conclusion part.

---

## Round 0.2 · accepted · Accept

You will be contacted by the PeerJ staff for further improving written English.